

**Study On The Driving Mechanism Of Hydrologic Drought In Karst**
**Basin Based On Landform Index: A Case Study of Guizhou, China**[*]
Zhonghua He [1,2*], Hong Liang [1], Zhaohui Yang [3], Xinbo Zeng [3]
1. School of Geographic and Environment Science, Guizhou Normal University, Guiyang, Guizhou, 550001, China
2.School of Geography and Environment Sciances, Guizhou Normal University / State Engineering Technology Institute for Karst
Desertfication Control, Guiyang, Guizhou, 550001, China
3. Department of Water Resources of Guizhou Province, Guiyang, Guizhou, 510275, China
**Abstract:** In recent years, hydrological droughts in the Karst Basins have become more frequent and have caused serious
ecological and environmental problems. This paper took the karst drainage basin of Guizhou, China as the study area to analyze the
geomorphologic distribution and the temporal-spatial variations of hydrological droughts. The results indicated that ① the rainfall
and its variation during drought periods had very limited impacts on the hydrological droughts in karst drainage basins; ② During
2000-2010, the hydrological droughts in Guizhou Province increased year by year, and the inter-annual variation of hydrological
droughts in Guizhou had obvious stage characteristics. The overall regional distribution of hydrological drought severity in Guizhou
is "*severe in the south and light in the north, severe in the west and light in the east*". ③ From the overall distribution of the landform
types, the mountains, hills and basins have a certain impact on hydrologic droughts, but the impacts are insignificant. From the
distribution of single landform types, the influences on hydrological droughts are particularly significant in high-medium mountains,
deep-high hills and high basins, and where are also relatively light areas for hydrologic drought severity, While the relatively serious
areas of that in the low mountains, shallow-low hills and low basins.
**Keywords:** Watershed Hydrological Drought; Geomorphologic Index; Landform Type; karst drainage basin

## 1. Introduction

In recent years, droughts have become more and more frequent, which, like large-scale disasters such as

floods, earthquakes and volcanic eruptions, are natural disasters that threaten human life and property security(EU,
2006, 2007; Sheffield et al., 2011). The nature of droughts is the lack of water in the basins. The main
source of basin recharge is atmospheric precipitation, followed by runoff recharge from the adjacent watershed
(for the karst watershed). The amount of recharge in the catchments is greatly affected by rainfall, and the impact
of basin topography on the primary distribution of precipitation should not be underestimated. In particular, the
landform types and its morphological characteristics, the combination of landform types and spatial features are
crucial to recharge / infiltration effect. Drought phenomenon is very complicated and has the characteristics of
temporal and spatial distribution as well as being influenced by human activities. Therefore, it is difficult to define
and study the drought simply (Van Loon et al., 2012).Therefore, the drought is usually divided into four types:

* Corresponding author.

[1,2]Zhonghua He (1976–); male; born in Xingyi, Guizhou; doctor, professor, and master tutor; and mainly engaged in the study on karst hydrology, water resource,
and remote sensing.



meteorological drought, agricultural drought, hydrologic drought and socio-economic drought (Van Huijgevoort et
al.,2014; Van Lanen et al.,2013). Hydrologic drought is the continuation and development of meteorological
drought and agricultural drought. It is the final and most complete drought that is caused by the river runoff below
its normal level due to imbalance between precipitation and surface water or groundwater(Dracup et al.,1980;
Feng, 1993).
The present studies on hydrologic droughts, the theory of runs is firstly applied to make quantitative
expressions for the characteristics of hydrological droughts (Yevjevich, 1967), and study the characteristics of
extreme hydrological droughts following the extremity of independent and dependent orders in normality, log
normality, and γ distribution (Sen,1977,1990, and 1991; Guven,1983; Sharma,1998). Utilizing the different
drought indices like the Regional Drought Area Index (*RDAI*) of daily runoff series and Drought Potential Index
(*DPI*) are to analyze the characteristics of regional hydrological droughts (Fleig ,2011), and study the relationship
of double variables between the drought duration and intensity (Kim,2006;Panu,2009). Employing the
Standardized Runoff and Rainfall Indexes (*SRRI*) are to study the influences of channel improvement and
nonlocal diversion on the process and level of hydrologic droughts (Wen, 2011). The level, process, and
recurrence interval of hydrologic droughts are studied by utilizing Palmer Drought Index (*PDI*), Soil Moisture
Model (*SMM*), Runoff Sequence (*RS*), Standardized Rainfall Index (*SRI*), and Vegetation Health Index (*VHI*),
respectively (Nyabeze,2004; Mondal,2015). Some scholars make a time series analysis and random simulation for
the hydrologic drought severity by using an autoregression model (Abebe, 2008), and make the Probabilistic
prediction of hydrologic drought by a conditional probability approach based on the meta-Gaussian model (Hao et
al.,2016), the seasonal forecasting of hydrologic droughts in the Limpopo Basin by a statistical analysis method,
respectively (Seibert et al.,2017). Rudd et al., (2017) was the first to use a national-scale gridded hydrologic
model to characterise droughts across Great Britain over the last century, and it was found that the model can very
well simulate low flows in many catchments across Great Britain. The threshold level method was also applied to
time series of monthly mean river flow and soil moisture to identify historic droughts (1891–2015), and it was
shown that the national-scale gridded output can be used to identify historic drought periods. Meantime, A small
number of scholars explore the spatial–temporal distribution differences between the characteristics of the
meteorological and hydrological droughts from the basin scale (Hisdal, 2003; Tallaksen, 2009). Among domestic
studies for hydrological droughts, the theory of runs is mainly applied to analyze the influence factors of runoff
volume in dry season and the identification of hydrological droughts (Feng,1997), and study the probability
density and distribution functions of extreme hydrological drought duration (Feng, 1993,1994, and 1995). Using
the fractal theory is to study the temporal fractal characteristics of hydrologic droughts, and estimate the
hydrologic drought severity by the time fractal dimension (Feng, 1997). Employing the Copula Joint Distribution
Function is to construct the joint distribution of hydrological drought characteristics (Zhou, 2011; Yan, 2007; Xu,
2010; Ma, 2010). However, most of the researches are still taking the different drought indices to make the
identification, characteristic analysis and prediction of hydrologic droughts, respectively. For example, Zhai et al.,
(2015) established a new hydrologic drought assessment index named Standard Water Resources Index (*SWRI*),
and developed a basic framework of hydrologic drought identification, assessment and characteristic analysis by
combining the distributed hydrologic model, Copula functions and statistical test methods. Zhao et al.,(2016)
selected the most suitable distribution from the logistic, normal, two-parameter log-normal, and Weibull



probability distributions to establish the Standardized Streamflow Drought Index (*SSDI*),classified the drought
magnitudes of hydrologic drought events by the *SDDI*, and validated the applicability and rationality of the *SSDI*
based on the actual drought situations in the Fenhe River Basin. Wu et al., (2016) constructed a Regional
Hydrologic Droughts Index (*RHDI*) combined with the percentages of runoff and precipitation anomalies,
obtained the frequency of corresponding drought grades, and then determined the threshold value of the different
drought grades based on the cumulative frequency of the *RHDI*. Tu et al., (2016) constructed the Copula Model of
two-variable joint distribution of hydrologic drought characteristics based on the test method of Cramer-von
Mises Statistics associated with Rosenblatt transfer, and analyzed the hydrologic drought characteristics under a
changing environment in Dongjiang River Basin. Based on the Variable Infiltration Capacity (*VIC*) model, Ren et
al., (2016) quantitatively separated the effects of climate change and human activities on runoff reduction, and
analyzed the spatial-temporal evolution characteristics of hydrologic droughts by the Standardized Runoff Index
(*SRI*). Li et al., (2016) analyzed the evaluation characteristics of the meteorological and hydrological droughts by
using Standard Precipitation Evapotranspiration Index (*SPEI*) and Streamflow Drought Drought Index (*SDI*), and
discussed the response of hydrological droughts to meteorological droughts. He et al., (2015) analyzed the
spatial-temporal characteristics of the meteorological and hydrologic droughts by Standardized Precipitation Index
(*SPI*), Standardized Discharge Index (*SDI*) and associated indicators with the trend, time lag cross-correlation
across the Yellow River Basin (YRB) during 1961-2010. Zhang et al., (2016) constructed the Copula prediction
model of hydrologic droughts based on the Copula Function and Runoff Distribution Function by the Standard
Runoff Index (*SRI*) according to the seasonal runoff-related characteristics, and made an empirical analysis for the
hydrologic station of the Aksu River West Bride.

However, the present studies on the hydrologic droughts in Karst basins, except for some relevant research

contents of this team (He et al., 2013, 2014, 2015,2018), have not seen a more detailed study reporting. Thus, this
paper is to take the Karst drainage basins in Guizhou Province of China as the study areas, make the identification
and quantification for hydrologic droughts by utilizing the Runoff Drought Severity Index (*RDSI*) (Feng, 1997 &
1997), and study the topographic features and hydrologic drought characteristics. And the driving mechanism of
hydrological droughts in karst basins is further studied.
## 2. Study areas
Guizhou Province, located in southwest China, adjoins Hunan Province to the east, Guangxi Province to the
south, Yunnan Province to the west and Sichuan Province and Chongqing Municipality to the north. Situated on
the east slope of the Yunnan-Guizhou plateau, it occupies an area of 176, 167 km$^2$ enclosed by coordinate points
of 24°37'N to 29°13'N, 103°36' E to 109°35'E (Fig. 1). The landscape in Guizhou is controlled deeply by the
geological structures, and is mainly dominated by basins, hills and mountains with towering mountains, cutting
strong, and significant elevation differences between valleys. Guizhou is an extremely developed karst province.
Karst topography is complete and widely distributed with the total area of the carbonate rock outcrops account for
73%. Guizhou Province is located in the subtropical East Asia monsoon region, and the climate type belongs to
China's subtropical humid monsoon climate. In most parts of the province, the climate is mild with no frost in
winter and no heat in summer, four distinct seasons, abundant annual rainfall and uneven spatial and temporal


distribution, and average annual precipitation across the province in the 1100~1300 mm. With poor lighting
conditions, lots of rainy days and high relative temperature, and 1200-1600 sunshine hours of every year in most
part of the province. The rivers in Guizhou are densely covered with a total length of 1,1270 km, of which 93 are
over 50 km in length. The Wumeng-MiaoLing Ridge watershed in Guizhou is a watershed, belonging to the
Yangtze River and Pearl River basins, ie the northern part of the Yangtze River Of the Jinsha River system, the
upper reaches of the Yangtze River mainstream system, the Wujiang River system and the Dongting Lake water
system, and the south of the Pearl River Basin Nanpanjiang River system, Beipanjiang River, Hongshuihe and
Duliujiang river system.


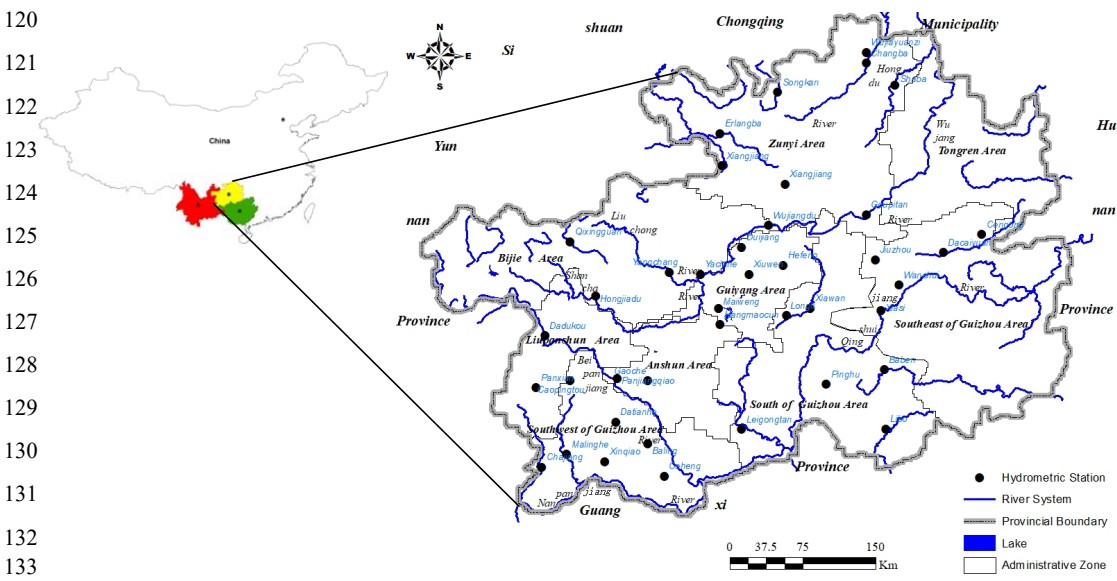

Fig.1 Sketch map of the study area
## 3. Data and methods
*3.1 Study data*
*(1) Hydrological data*
Considering the typicality and representativeness of hydrological data and the continuity and homogeneity of
the hydrological data in this study area, this paper selected the monthly runoff and rainfall measurements of 40
hydrometric stations in Guizhou Province (Fig. 1). Hydrological data were collected from "*Guizhou Statistics on*
*Mean Monthly Flows per Calendar Year*" compiled by *Guizhou hydrologic station*, with reference to "*Guizhou*
*Water Resource Report*" compiled by *Guizhou Hydrology & Water Resources Department*, and selected annual
minimum monthly average runoff and the average monthly rainfall with the time range from January 2000 to
December 2010 .
*(2) Remote sensing data*
Taking into account the evolution of the geomorphology is a slow and long geological process, and the type




and shape of the topography in 2000-2010 remained basically unchanged. Therefore , this paper extracted the
geomorphological information based on the LS5_TM images of the month corresponding to the minimum
monthly mean runoff in 2006 (Time: January to December 2006; Strip Number & Line Number: 126~129,
040~043; Data Format & Level: ∗∗.geotiff, L4).The Digital Elevation Model (DEM) is based on data provided by
the United States Geological Survey (USGS)(Data Format: Grid; Coordinate System: WGS_84; Spatial
Resolution: 30 m).
*3.2 Study methods*
*(1) Identification of hydrological drought*
Hydrological drought is the phenomenon when the river flow is lower than its normal value. In other words,
the river flow cannot satisfy the water supply demand in a certain period (Van Loon et al., 2012, 2015; Mishra,
2010). The run theory (Herbst et al., 1996) was adopted to identify hydrologic drought (Fig.2). For a runoff time
series $x(t)$, a significant drought period could be taken as $X(t) < X_0(t)$ after applying a truncation level $X_0(t)$. The
length of negative runs $D(X(t)<X_0(t))$ is the duration of drought L. The total number of negative runs is the total
deficit of water for the drought S. The intensity of negative runs is the magnitude of drought M, indicating the
average water deficit volume of the drought period: $M=S/L$.

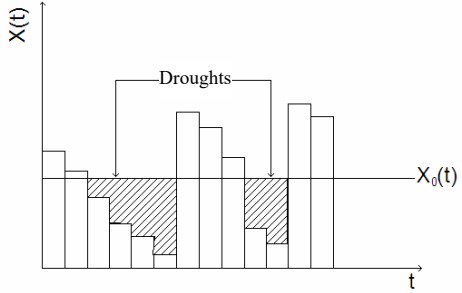

**Fig. 2** Identification of hydrologic droughts
In this paper, hydrologic droughts in the karst drainage basins were identified by using the Mean Monthly
Flow (*MMF*) of the period from 2000 to 2010 as truncation level. And taking the *MMMF* of sampling sites as *Y*
axis and the series of sampling sites as *X* axis. Because of Hydrologic Drought Severity (*HDS*) mainly depends on
the volume of water deficit and the length of drought duration, this paper took Relative Drought Severity Index
(*RDSI*) (Feng et al., 1997) as the measurement of *HDS*, and the formula for calculating *RDSI* was presented in the
equation below:
$$RDSI = LD \times DI \tag{1}$$
Where *LD* is the relative drought duration within a year; (valued as 1/12 in this paper). *DI* is the relative
water deficit of that drought period.
To eliminate the impact of units of measurement for the runoff, the following non-dimensionalization
equation was adopted:



$$DI = \frac{X_i - X_{mean}}{X_{mean}} \qquad (2)$$

Where $X_i$ is the *MMMF* for sample site *i*; $X_{mean}$ is the *MMF*, viz., the truncation level.
*RDSI* is a negative value and the larger the absolute value is, the more severe the drought is.
(2)*Landform index*
This paper made some processes on the spectral radiance and apparent reflectance of remote sensing data
corresponding to the minimum monthly average runoff depth of the hydrological station in 2006, and extracted the
sample sites controlled by the hydrologic cross-section (He et al., 2012). The object-oriented classification
technology was been used to extract the Geomorphic Type Indicators (*GTI*) and Landform Index (*LI*)   based on
the *GTI* and *LI* (Tab.1 and Tab.2) (MA et al., 2012), and referring   to "*Guizhou Geomorphology Map*" (internal
data) compiled by Guizhou Normal University.
Tab. 1    Basic classification of landforms

| 1st grade landforms | 1st grade classification. criteria Depth of dissection, surface D(m) | 2nd grade landforms | 2nd grade classification criteria Absolute altitude H(m) | 3rd grade landforms | 3rd grade classification criteria Depth of dissection, surface D(m) |
|---|---|---|---|---|---|
| Basin | S < 9° | Depression | Slope of basin bottom  < 5° and area < 1 km² | | |
| | | Low | H < 900 | | |
| | D < 100 | Medium | 900≤H < 1900 | | |
| | | High | 1900≤H | | |
| Hill | 9°≤s < 14° | Low | H < 900 | Shallow | D < 200 |
| | | | | Deep | 200≤D |
| | | Medium | 900≤H < 1900 | Shallow | D < 200 |
| | | | | Deep | 200≤D |
| | | High | 1900≤H | Shallow | D < 200 |
| | | | | Deep | 200≤D |
| Mountain | 14°≤S | Low | H < 900 | | |
| | | Medium | 900≤H | Low | 900≤H < 1400 |
| | | | | Mid | 1400≤H < 1900 |
| | | | | High | 1900≤H |


Tab. 2   Indices for landform classification

| Name | Formula | Range | Description |
|---|---|---|---|
| symmetry | $\dfrac{2\sqrt{\frac{1}{4}(VarX + VarY)^2 + (VarXY)^2 - VarX VarY}}{VarX + VarY}$ | [0,1] | VarX: Variance in   X direction VarY: variance in Y direction. Eigenvalue rises with symmetry |
| Square fit index (or density index) | $\dfrac{\sqrt{\#P_v}}{1 + \sqrt{VarX + VarY}}$ | [0, a value determined by the shape of image object] | $\sqrt{\#P_v}$ : diameter of square object containing $\#P_v$ pixel. $\sqrt{VarX + VarY}$ : diameter of the ellipse $P_v$ : image object V expressed in pixels The more the image object resembles a rectangle in shape the higher its characteristic value, |
| Rectangle fit index | $\dfrac{\#\{(x,y) \in P_v : \rho_v(x,y) \le 1\}}{\#P_v} - 1$ | [0,1]. 1: 100% fit, 0 : 0% of pixels fit into the rectangle | $\rho_v(x,y)$: rectangular distance at a pixel (x,y). |
| Ellipse fit index | $2 \cdot \dfrac{\#\{(x,y) \in P_v : \varepsilon_v(x,y) \le 1\}}{\#P_v} - 1$ | [0, 1], 1: 100% fit, 0: ≤ 50% of pixels fit into the ellipse. | $\varepsilon_v(x,y)$ :ellipse distance at a pixel (x,y). $P_v$ : image object V expressed in pixels $\#P_v$ : image object V expressed in pixels |

**Note:** *Definiens Developer7 Reference Book was consulted for this index*




## 4. Results and analysis

*4.1. Geomorphic distribution characteristics of Karst basins*

*4.1.1 Distribution characteristics of geomorphic types*

The overall landscape of Guizhou is dominated by mountains, followed by hills and basins. And these mountains ware mostly dominated by low-medium and mid-medium mountains with the total area of the province accounting for 27.37% and 16.94%, respectively, followed by low mountains (10.96%) and high-medium mountains (4.93%). Hills are dominated by low hills with an area of 22.06%, followed by mid-hills (9%) and high hills (3.09%). Basins ware mostly low basins (4.86%), followed by medium basins (0.51%), high basins (0.25%) and a few depressions (0.012%). Guizhou is a mountainous province with mountainous areas all over the province, while only a few areas are less widely distributed, such as Liupanshui and Anshun areas, but a large proportion of hilly areas are distributed in Guizhou, and mountains and hills in Guizhou show a "symmetrical" distribution. Hilly landforms in the province are distributed, but presented "trough" in the Southwest area, "broken" phenomenon in Zunyi. Basins are less distributed in the whole province, and presented "broken" phenomenon in the parts of southwest area, Liupanshui, Anshun and Zunyi (Fig. 3a).

*4.1.2 Characteristics of topographic relief degrees*

In Guizhou, the spatial distribution of the Topographic Relief Degrees (*TRD*) of mountains is basically consistent with that of the hills, and the peak *TRD* of mountains presents in Dadukou, Shuicheng (relative relief 1898m), Panxian (relative relief 1885 m) and Chajiang, Xinyi (relative relief 1842 m). While the maximum of relative relief of hills presents in Maiweng, Pingba (1518 m), and Hefeng, Kaiyang(896.4 m)and Xiawan, Guidiing (870.28 m)(Fig. 3b).

*4.1.3 Distribution characteristics of landforms*

From the analyses of the symmetry of topographic distribution, the symmetry indices of the three types of landforms all fluctuate around 0.6, indicating that there is a certain degree of "*symmetry*" in the mountain landform, hilly landform and basin topography (Figure 3c), and the symmetry index of mountain topography fluctuate within 0.4 ~ 0.8, the hills fluctuate within 0.2 ~ 0.9, and the basins fluctuate within 0.4 ~ 1. The square fitting index (density index) of the mountains, hills and basins all fluctuate around 1.5, indicating the "*squareness*" distribution of the topography of the mountains, hills and basins. In general, the hilly square fitting index (density index) is greater than the mountain, indicating that the hilly landform morphology is closer to "*square*" than the mountain topography (Fig 3d). The rectangle fitting index of hilly landform is generally greater than that of mountainous area, and the rectangle fitting index of mountain topography fluctuates within 0.4 ~ 0.7 (Fig. 3e). Similarly, the elliptic fitting index of the hilly landform is generally greater than that of the mountainous area. The elliptic fitted index of the hilly and basin fluctuates greatly, ie., varies from 0 to 0.6 and from 0 to 0.7, respectively, and the "*broken*" phenomenon occurs in some areas (Fig.3f) .

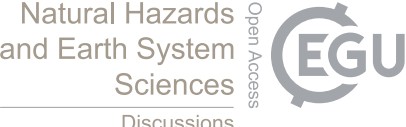



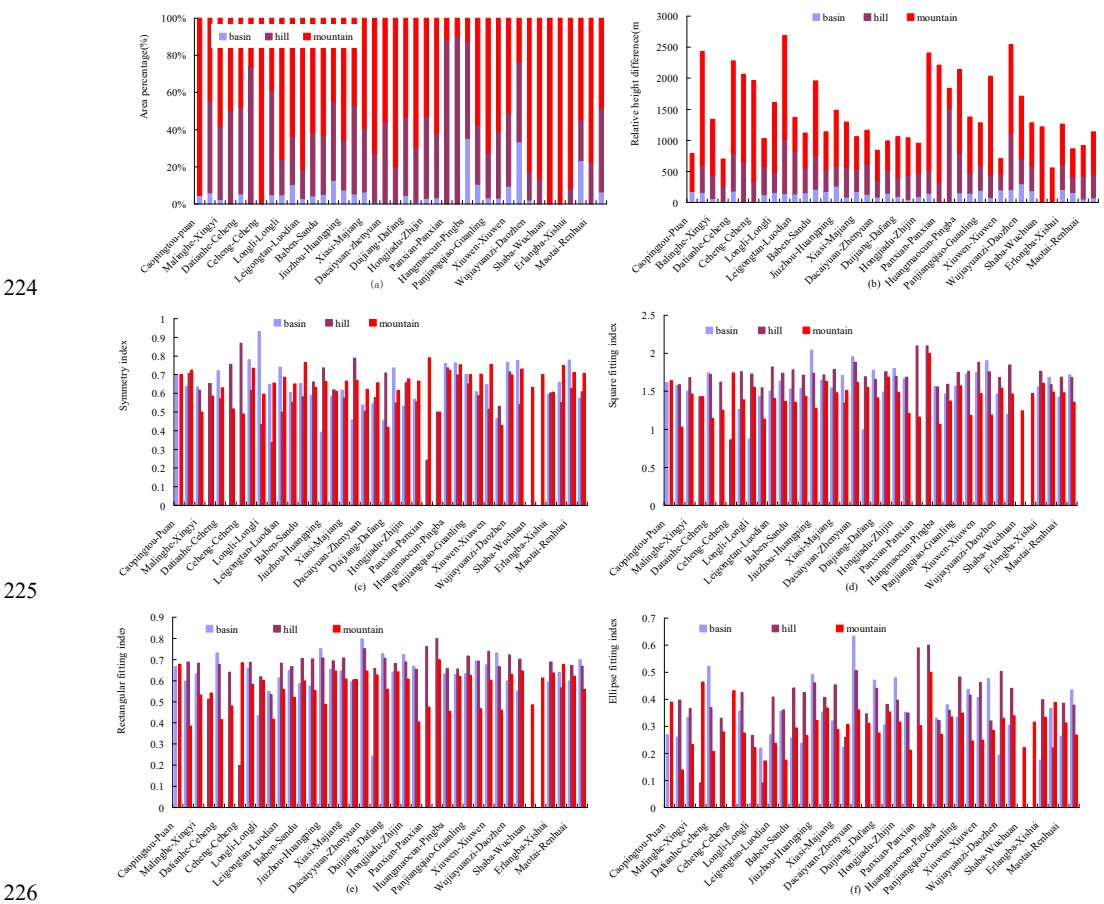




Fig.3 The overall distribution of landform types

### 4.2. Hydrologic drought characteristics in Karst basins

#### 4.2.1 Inter-annual variation characteristics of hydrological drought

During 2000-2010, the hydrological droughts in Guizhou Province increased year by year, most notably in 2010 ($RDSI$ = -0.634), followed by 2005 ($RDSI$ =-0.591) and 2009 ($RDSI$=-0.555), the lighter was in 2000 ($RDSI$ = -0.528). From 2000 to 2010, the inter-annual variation of hydrological droughts in Guizhou had obvious stage characteristics, which could be generally divided into "*three stages and four periods*", that was, the first transitional period from 2000 to 2001 (relative annual rate was 10.13% ), 2004-2005 as the second transitional phase (annual relative variability of 11.09%), 2009-2010 for the third transitional phase (annual relative variability of 18.76%), and 2000 for the first period of drought, 2004 for the second period of drought, 2005-2009 for the third period of drought, 2010 for the fourth period of drought (Fig.4a).

During 2000-2010, the coefficient of variation ($C_v$) of hydrological droughts in Guizhou showed obvious inter-annual variability, showing a tendency of decreasing year by year. The inter-annual variation of hydrological droughts occurred most frequently in 2000 ($C_v$=-0.685) and in 2004 ($C_v$=-0.65), with relatively small inter-annual





variations in 2010 ($C_v$=-0.385) and 2001 ($C_v$=-0.487). The inter-annual differences of the $C_v$ values of regional
hydrological droughts was significant with annual relative variability as high as 66.11% (2000-2001), followed by
2009-2010 (relative annual rate of 51.04%), 2004-2005 (rate of 30.94%). The *RDSI* of hydrological droughts was
opposite to the $C_v$ of hydrological droughts, that was, the greater the *RDSI* value of hydrological droughts, the
smaller the $C_v$ value of hydrological droughts (2010). On the contrary, the smaller the *RDSI* value of hydrological
droughts was, the greater the $C_v$ value of hydrological droughts (2000) was. The inter-annual variation trends of
the *RDSI* and $C_v$ values of hydrological droughts was the opposite (Fig.4a).

*4.2.2 Spatial distribution of hydrologic drought*

The overall regional distribution of hydrological drought severity in Guizhou is "*severe in the south and light*
*in the north, severe in the west and light in the east*" (Fig. 4b). The most severe areas of hydrological drought
appeared in the "*Southwest Guizhou Province*", and the relatively light areas of that in the "*Zunyi Area*". The
regional variation of $C_v$ values of hydrological droughts is divided into two sections, that is, the first half is
"*curved- type*" and the second half is "*W-shaped* ", which shows the regional variation of $C_v$ values is small in the
southern part of Guizhou, and large in the other areas. For example, the $C_v$ value of hydrological drought in
Liupanshui reaches a maximum value ($C_v$=-1.595), and the $C_v$ value of hydrological drought in Zunyi reaches a
minimum ($C_v$=0.207).The Northeast Southwest Distribution (Fig. 4c): hydrological drought severities "gradually
increased", and showed a small "*wave-type*" distribution. The regional variation of $C_v$ values is greatly, and shows
"*N-type*" distribution. The Northwest Southeast Distribution (Fig. 4d), North-South Distribution (Fig.4e) and
Western Distribution (Fig.4f): the *RDSI* values of hydrological droughts in Karst basins are both greater than
-0.44.The hydrologic drought severities gradually increase, showing a "linear" distribution with linear fitting
indices $R^2$=0.995, $R^2$ = 0.9978 and $R^2$=0.3794, respectively. The $C_v$ values of regional hydrological droughts vary
greatly, showing a "*V-shaped*" distribution. The Southern Distribution (Fig.4g): the hydrological drought severities
in Karst basins ware "gradually reduced" with a "linear" distribution ($R^2$=0.9633), and the regional variation of $C_v$
values of hydrologic droughts is small.

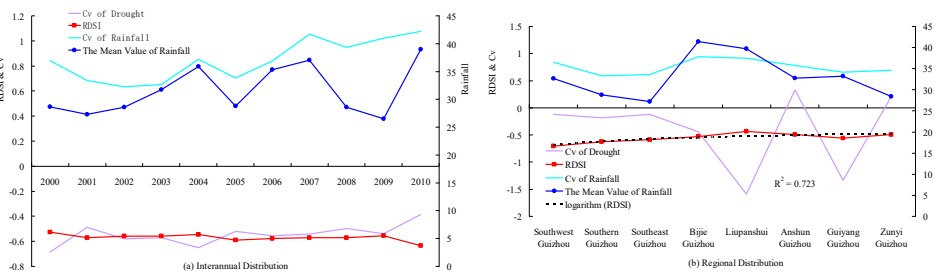




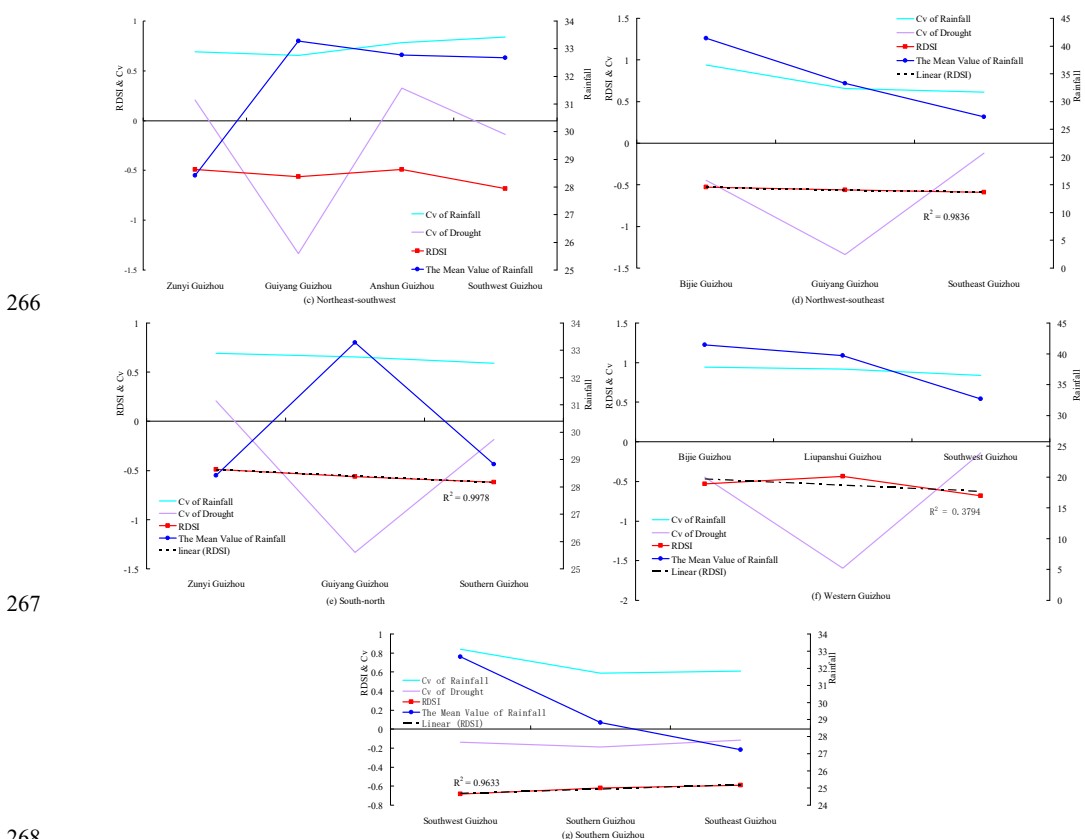




Fig.4    The spatial and temporal distribution of hydrological droughts and impacts of rainfall factors
*4.3. Driving mechanism of hydrologic drought in Karst Basins*
*4.3.1 Driving mechanism of rainfall factors to hydrologic drought*

*(1) Inter-annual changes driven by rainfall factors*

Watershed hydrological drought refers to the phenomenon of shortage of water due to different basin

underlying factors when rainfall is low (or constant). Feng (et al.,1997) Pointed out that the runoff in the dry
season mainly comes from the amount of water retained in the catchment at the end of the flood season, and the
amount of rainfall in the dry season, with the former accounting for a large proportion. The flood storage at the
end of the flood season is mainly determined by the flood season precipitation and catchment factors, and the
latter represents a significant proportion. As can be seen from Fig.4a, the mean value of rainfall in the driest
month was "increasing year by year" from 2000 to 2010, while the hydrological drought severity in Karst basins
was "serious year by year", which indicates that rainfall during drought periods has little effect on hydrological
drought with the correlation coefficient $R$=-0.468, and significant probability $P$=0.147. In 2010, the average
rainfall of the driest month was 38.949 mm with the *RDSI*=-0.634, and 28.651 mm with the *RDSI*=-0.528 in
2000.The difference of average rainfall of the driest month in 2001-2004 and 2005-2009 was very big, but that of
drought degree was very small. In 2000-2010, the inter-annual variability of $C_v$ values of the average rainfall in





the driest month was great, and showed an "increasing" trend (Fig.4a), while that of $C_v$ values of hydrologic
droughts was relatively small, and showed an "decreasing" trend, which indicates that the change of rainfall in the
driest month has little effect on the hydrologic drought severity with the $R$=0.323, $P$=0.332. Similarly, the $C_v$ value
of the average rainfall in the driest month was 1.075, the drought index $C_v$=-0.385 in 2010, and the average
rainfall $C_v$=0.843, the drought index $C_v$=-0.685 in 2000. Similarly, the $C_v$ value of the average rainfall in the driest
month was 1.075, the drought index $C_v$=-0.385 in 2010, and the average rainfall $C_v$=0.843, the drought index
$C_v$=-0.685 in 2000. From 2005 to 2008, the great was the Inter-annual variability of $C_v$ values of the average
rainfall in driest month, the small that of $C_v$ values of hydrologic drought severities.
*(2) Regional changes driven by rainfall factors*
In Guizhou Province, the spatial distribution of the mean rainfall in the driest month changed greatly, with a
"hump type" (Fig.4b).The spatial distribution of the *RDSI*s has little change and has a "logarithmic" distribution
with a logarithmic fitting index of $R^2$=0.723. This indicates that the spatial distribution of rainfall in the driest
month has little effect on that of the *RDSI*s, and Pearson correlation coefficient $R$=0.4, the significant probability
$P$=0.326. The $C_v$ of rainfall has little effect on the $C_v$ of hydrological droughts ($R$=-0.27, $P$=0.518).*The Northeast*
*Southwest Distribution* (Fig.4c) and *North-South Distribution* (Fig.4e): the spatial distribution of rainfall in the
driest month changes a lot and appears "single peak". The rainfall had little effect on the watershed hydrological
droughts. The correlation coefficient and significance ware $R$=-0.454, $P$=0.546, and $R$= -0.122, $P$=0.922,
respectively. The space change of the $C_v$ of rainfall is small, which has a small influence on the $C_v$ of hydrological
droughts. The correlation coefficient and significance ware $R$=-0.55, $P$=0.45, and $R$=0.87, $P$=0.945, respectively.
*The Western Distribution* (Fig.4f): The spatial variation of rainfall is small and shows a "decreasing" trend. The
rainfall has no significant effect on hydrological droughts ($R$=0.841, $P$=0.364). There is no linear correlation
between the $C_v$ of rainfall and the $C_v$ of hydrological droughts ($R$=-0.478, $P$=0.683). *The Northwest southeast*
*distribution* (Fig.4d) *and southern distribution* (Fig.4g): The rainfall dropps drastically, and has a significant
impact on hydrological droughts, the correlation coefficient and significance ware $R$=0.998, $P$=0.041, and
$R$=-0.999, $P$=0.028, respectively. However, the $C_v$ of rainfall has no significant effect on the $C_v$ of hydrological
droughts, and the correlation coefficient and significance ware $R$=0.135, $P$=0.913, and $R$=0.302, $P$=0.805,
respectively.
*4.3.2 Driving mechanism of landforms characteristics to hydrologic drought*
*(1) The driven by landform types*
During the drought period, there is no rainfall or little rainfall in the karst catchments, which could not solve
the drought problem. The runoff recharge mainly comes from the rainfall at the end of the flood season, and the
recharge in the adjacent catchment (the non-closed catchment).Therefore, the topography type plays an important
role in rainfall recharge. Different types of landforms, such as landforms, topographic relief degrees (Ma et al.,
2012) and surface cutting depths of them are quite different, greatly influence on the horizontal flow on the
surface and vertical flow under the ground of the rainfall, affect the rainfall recharge to the basin, and which relate



to the occurrence of watershed hydrological droughts. From the overall geomorphologic types of Guizhou, the
area distributions of mountains, hills and basins are related to *RDSI*. The correlation coefficients are
$R_{(mountain)}$=-0.399, $R_{(hill)}$=-0.212 and $R_{(basin)}$=0.209, respectively. Except basins, Hills and mountains do not pass the
significance test of 0.05. From the correlation between single landform types and *RDSI* (Fig. 5a), the correlation
could be divided into three sections. They are the basin section, showing "*N type*" and hilly section, showing
"*bimodal type*" and mountainous section, showing "*growth type*". In the basin section, the smallest is the
correlation between low-lying basins and *RDSIs* (*R*=-0.291, *P*=0.069), the highest in the high basins (*R*= 0.478,
*P*=0.002). In the hilly section, the smallest is the correlation between shallow low hills and *RDSIs* (*R*=-0.241,
*P*=0.134), the highest in the deep high hills (*R*=0.523,*P*=0.001), followed by deep-medium hills(*R*=0.177,
*P*=0.273). In the mountainous section, the highest is the correlation between high-medium mountains and *RDSIs*
(*R*=0.414,*P*=0.008), the smallest in the low mountains(*R*=-0.073,*P*=0.653).The *RDSI* value of hydrological
droughts is negative. That is, the greater the negative, the more severe the hydrological drought severity. Therefore,
the correlation coefficients (*Rs*) of the landform types and *RDSIs* are larger, the more significant the influences of
the landform types on the hydrological droughts are, and the lighter the hydrological droughts are. On the contrary,
the greater the negative *Rs* between the landform types and the *RDSI*s, the more significant the influences of
topography on hydrological droughts are, and the more serious the hydrological droughts are. In summary, the
correlation coefficients (*Rs*) of high-medium mountains, deep-high hills and high basins are all greater than 0 and
through the significance test of 0.01, which indicates that it is the relatively lightly areas of hydrologic droughts in
the high-medium mountains, deep-high hills and high basins, while the relatively serious areas in low basins,
shallow-low hills and low mountains with the negative *Rs*. With the elevation increasing, the *Rs* between landform
types and *RDSIs* change from negative to positive and then increase in the basins, hills and mountains, which
indicates that it is getting lighter for the watershed hydrological droughts with the altitude increasing. This could
be that the high altitude area has low erosion basis and shallow groundwater, while the low altitude area would
have the opposite situations.
*(2) The driven by landform dissection depths*
Affect the lateral velocity of surface water produced by rainfall, in addition to landform types, its relief
amplitude or the depth of dissection could not be underestimated. The deeper the surface-cutted depth, the greater
the surface fluctuation (correlation coefficient $R_{basin \& basin}$=0.842, $R_{hill \& hill}$=0.982 and $R_{mountain \& mountain}$=0.362), and
the longer the confluence of rainfall on the surface, the more rainfall infiltration, so the lighter the ydrological
drought severity occurs. As shown in Fig. 5a , the correlation between surface cutting depth and *RDSI* could be
divided into three sections, that is, the basin section is "*V-shaped*" and the hilly section is "*W- type*" and the
mountain section is " *V-type* ".
Similarly, the impacts of the surface-cutted depths in high basins, deep-high hills and high-medium
mountains on hydrologic droughts are the largest with the correlation and significance of *R*=0.536 and *P*=0.0,
*R*=0.568 and *P*=0.0, *R*=0.557 and *P*=0.0, respectively. while those of low basins, deep-low hills and low-medium
mountains are the smallest with the *R*=0.148 and *P*=0.361,*R*=-0.092 and *P*=0.572,*R*=-0.104 and *P*=0.522,


respectively. This indicates that  the relatively light areas for hydrologic drought severity in the high basins,
deep-high hills and high-medium mountains, and the relatively serious areas in low basins, deep-low hills and
low-medium mountains, which could be because that deeper dissection provides more time for the rainfall to form
surface flows and increases the volume of infiltration. The *R* between *RDSI* and surface-cutted depth from
depression to high-medium mountain is generally "increasing", which shows that the hydrologic drought severity
in Karst basins is a getting lighter trend from depression to high-medium mountain.
*(3) The driven by landform characteristics*
Another important characteristic value of geomorphology types is morphological index, such as symmetry
index, square fitting index (density index), rectangle fitting index and ellipse fitting index, which reflect the
morphological characteristics and shape complexities of landform types from a different point of view, and also
reflect the closure degree of the surface-groundwater system. Fig.5b is the correlation between morphological
index and *RDSIs*, similarly divided into three sections. That is, a "*V type*" for the symmetry index, square fitting
index and rectangular fitting index, a "*N type*" for the ellipse fitting index in basin sections, and a "*U-shaped*" for
the symmetry index, square fitting index and rectangular fitting index, a "*W-type*" for the ellipse fitting index in
hilly sections, and a "*V-shaped*" for the four kinds of morphological index in mountain sections. The *Rs* between
four kinds of morphological indices of the high basins, deep-high hills and high-medium mountains and the *RDSIs*
are greater than 0, and *P*=0.0, which indicates that it has a significant impact on hydrological droughts, and is also
a relatively light area for hydrologic drought severity in the high basins, deep-high hills and high-medium
mountains. The *Rs* between four kinds of morphological indices of the mid-medium mountains and the *RDSIs* are
the minimum, which shows that the shape distribution of mid-medium mountains has no obvious or no influence
on the watershed hydrological droughts. From depressions to high mountains, the *Rs* between the four
morphological indices and *RDSIs* are positive (except the low-medium mountain by ellipse fit index). Especially
from depression to deep-high hills, the *Rs* between of them are the relatively large, which indicates that the
morphological distribution of landform types has different impact on watershed hydrologic droughts. It could be
that the larger the morphological index of morphological types, the more regular the shape distribution of the
landscape, and the simpler the edge distribution of landform types, the less outflow of water out of the basin, and
the smaller the probability of watershed hydrological drought occurs.

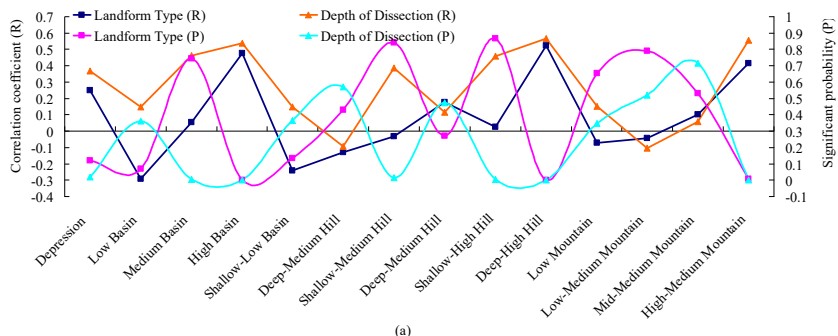

(a)




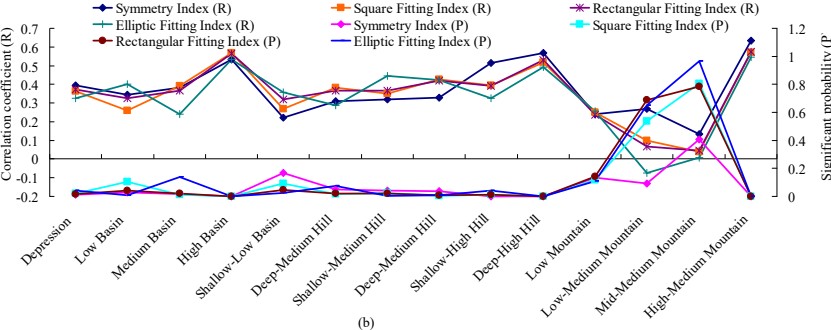

(b)

Fig.5 The correlation coefficients between landform types and hydrological droughts

### 4.3.3 Driving mechanism of hydrologic drought variability

*(1) The inter-annual variability driven of hydrologic drought*

Hydrological drought is the continuation and development of meteorological drought and agricultural drought. It is the final and most complete drought. However, once hydrological drought has occurred, it indicates that ① the deficiency of rainfall has reached an abnormal level; ② the catchment has a water deficit and a lower groundwater table; ③ irrigation is no longer possible (Geng et al., 1992).Once the hydrological drought occurs, it will be a devastating damage to the ecological environment of Karst basins. The result shows that the soil water content or soil water holding capacity drops sharply, even reaches the level of withering coefficient, which makes it difficult to supplement plant physiological water demand, resulting in the drying up and death of large areas of crops and vegetation. In addition to that, the vegetation coverage will drop, making the soils and rocks of the catchment become naked and scorched and thereby producing a lot of sands and dusts that may aggravate the greenhouse effect. Water storage medium is seriously damaged, thus affecting the basin's water storage capacity, which is an important factor that led to the hydrological drought in the coming year. Making pairwise correlation analysis on the *RDSI*s of 2000-2010 showed that each correlation coefficient was above 0.501, each significance probability was below 0.001, which indicates that the inter-annual affect each other of hydrologic droughts during 2000-2010 was particularly significant.

*(2) The regional variability driven of hydrologic drought*

The Karst drainage basin is a binary three-dimensional space structure with the binary media and dual water systems. According to the closed degree of surface water system and groundwater system, YANG (1982) classified karst basins into the Surplus Basin, Balanced river Basin and Deficit Basin. On the one hand, in karst basins, the basin water storage in the dry season is the main source of basin runoff recharge. Therefore, the strong / weak of basin water storage capacity affects the amount of runoff during the dry season, which is directly related to the occurrence of hydrological droughts. On the other hand, the water storage capacity of karst basins is greatly influenced by the basin storage medium and its water system. The water storage medium affects the type of water storage space, the size of water storage space and the numbers of water storage space, which will affect the amount of basin water storage. Water system is the channel of energy flow and information flow, which is the reflection of secondary distribution of rainfall on the surface and is the key factor of water balance in the basins.



In the karst areas, the rainfall during the dry period has little effect on the surface runoff. The runoff recharges
mainly come from the water storage in the basin or the water storage in the adjacent basin. Therefore, it is
significant for the mutual influence of hydrologic droughts in the Adjacent drainage basins, for example, in the
Bijie area & Guiyang City ($R$=0.832, $P$=0.01), Bijie area & Anshun area ($R$=0.816, $P$=0.014), and Anshun area &
Guiyang city ($R$=0.753, $P$= 0.031). However, they may belong to neighboring areas from the administrative
divisions, which could be that the surface water system and the groundwater system are not closed, resulting in the
exchange of groundwater. If there is no exchange of groundwater or  has not been lost of surface water,
hydrologic droughts will have little or no influence on each other even in the adjacent areas, such as Qianxinan
area & Anshun area ($R$=-0.199, $P$=0.637).
**5. Conclusions**

Based on the TM images and DEM data, This paper extracted the landform types, the morphological indices

of geomorphology types, the topographic relief degrees and so on by the use of the object-oriented classification
technique, and systematically analyzed the distribution of geomorphology types in Guizhou, the temporal and
spatial distribution of hydrological droughts in Karst basins, and the correlation between the rainfall during dry
periods, geomorphology types and the hydrological droughts in the basins. The results show that:

(1)During 2000-2010, the hydrological droughts in Guizhou Province increased year by year, most notably

in 2010 ($RDSI$=-0.634), which was in line with the southwestern drought in 2010. The inter-annual variation of
hydrological droughts had obvious stage characteristics, which could be generally divided into "three stages and
four periods", that was, the first transitional period from 2000 to 2001 (relative annual rate of change was
10.126%), the second transitional period from 2004 to 2005 (relative annual rate of 11.01%),  and the third
transitional period from 2009 to 2010 (relative annual rate of 18.76%). 2000 was the first drought period,
2001-2004 was the second drought period, the third period of drought in 2005-2009 and the fourth period of
drought in 2010. The overall regional distribution of hydrological drought severity in Guizhou was "*severe in the*
*south and light in the north, severe in the west and light in the east*". The most severe areas for hydrological
drought severity appeared in the "*Southwest Guizhou Province*", and the relatively light areas for that in the
"*Zunyi Area*".

(2) The rainfall during drought periods has little effect on hydrological drought. For example, the mean value

of rainfall in the driest month was "increasing year by year" from 2000 to 2010, while the severity of hydrological
droughts in Karst basins was "serious year by year". The change of rainfall in the driest month has little effect on
the severity of hydrologic droughts. For example, in 2000-2010, the inter-annual variability of $C_v$ values of the
average rainfall in the driest month was great, and showed an "increasing" trend, while that of $C_v$ values of
hydrologic droughts was relatively small, and showed an "decreasing" trend. The spatial distribution of rainfall in
the driest month has little effect on that of the $RDSI$s. For example, the spatial distribution of the mean rainfall of
the driest month in Guizhou Province showed a great change with a "hump type" distribution. The spatial
distribution of the $RDSI$s showed a small change with a "logarithmic" distribution.



(3) During the dry period, it is significant for the mutual influence of hydrologic droughts in the Adjacent
drainage basins, for example, in the Bijie area & Guiyang City (*R*=0.832, *P*=0.01), Bijie area & Anshun area
(*R*=0.816, *P*=0.014), and Anshun area & Guiyang city (*R*=0.753, *P*= 0.031).This may be that the rainfall during
the dry period has little effect on the surface runoff in the karst areas, and the runoff recharges mainly come from
the water storage in the basin or the water storage in the adjacent basins. At the same time, the inter-annual affect
each other of hydrologic droughts during 2000-2010 was particularly significant.
(4) From the overall geomorphologic types of Guizhou, the area distributions of mountains, hills and basins
have certain influence on the hydrological droughts in Karst basins, but the effect is not significant. From the
distributions of single landform types, the influence of high-medium mountains, deep-high hills and high basins
on hydrological droughts is especially significant. And it is relatively light area for hydrologic droughts in the
high-medium mountains, deep-high hills and high basins, and is relatively serious area in low basins, shallow-low
hills and low mountains. This indicates that the hydrological droughts in Karst basins are the more and more light
with the altitude increasing. The correlations between depth of dissection and *RDSI* from depression to
high-medium mountain are generally "increasing", which indicates that the hydrologic droughts in the basins
show a tendency of "getting lighter and lighter". There are significant impacts on the hydrological droughts for the
landforms distribution of   high basins, deep-high hills and high-medium mountains, and where are also relatively
light distribution areas of hydrologic drought severity. From depressions to high mountains, the correlation
coefficients (*Rs*) between the four morphological indices and *RDSIs* are positive (except the low-medium
mountain by ellipse fit index), and the relatively large for the *Rs* especially from depression to deep-high hills,
which indicates that the morphological distribution of landform types has different impact on hydrologic droughts
in Karst basins. This could be that the larger the morphological index of morphological types, the more regular the
shape distribution of the landscape, and the simpler the edge distribution of landform types, the less outflow of
water out of the basin, and the smaller the probability of hydrological droughts in Karst basins occurs.


## Acknowledgements

The authors are grateful to the Editors and the anonymousreviewers for their useful suggestions and
comments. This study was supported by the Natural Science Foundation of China (41471032; u1612441); Project
for National Top Discipline Construction of Guizhou Province (85 2017 Qianjiao Keyan Fa);
Project of National Key Innovation Base Construction      (Qiankeheji Lab [2011]4001);Natural and scientific
research fund of Guizhou Water Resources Department (KT201402); Natural and scientific fund of Guizhou
Science and Technology Agency (QKH J [2010] No. 2026, QKH J [2013] No. 2208); 2015 Doctor Scientific
Research Startup Project of Guizhou Normal University.






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
