# Peer review of "Study On The Driving Mechanism Of Hydrologic Drought In Karst"

_Natural Hazards and Earth System Sciences, 2018_

## Referee Comment (RC1) · Anonymous Referee #1 · 1 Jun 2018

This paper is focused on spatial and temporal variations of hydrological droughts using a standardized runoff index and geomorphologic indices for the karst drainage basin of Guizhou, China. The paper is a new research study and is generally well-written as it explains the methodology, the mathematical framework and the assumptions used. However, the application research part needs improvements and corrections to verify the novelties of the method employed in the study area. Based on this general comment the following points should be addressed and clarified.

1. Threshold Selection. Please provide information on threshold selection. Why the authors select a fixed threshold (Mean Monthly Flow, MMF if I understood correctly).

Why a variable threshold method is not selected for this study (e.g. Van Loon, 2015)? I would expect from the authors to use a monthly varying threshold for this type of presented analysis. Please justify this issue on the revised manuscript.

2. Regional Analysis (Page 5 and 6 of the manuscript). Please explain how the regional analysis is performed? The authors use the MMMF index? "And taking the MMMF of sampling sites as Y axis and the series of sampling sites as X axis". Is MMMF a regional index and how is derived? Is it the mean? Do you think that the mean index is representative considering the small dataset of 10 years? I would suggest to use an unbiased index if of course is it possible (maybe the median of the sites?). Please address this issue on the revised manuscript.

3. Standardization procedure of runoff index. (Equation 2 application). Please explain in detail the standardization procedure of the DI (the relative water deficit). Equation 2 is valid only when the variables are close to nornal distribution. This should be addressed in the revised manuscript. For example several theoretical distributions could be tested on DI values or if the data do not follow nornal distribution normalization techniques could be followed (i.e. Box-Cox transformation in Vasiliades et al., 2011). Furthermore, please take into account in your analysis the small temporal dataset in your analysis. Usually >30 years are needed to derive drought indices. Therefore, I am quite skeptical with the use of the term with the mean in Equation 2. I would recommend to the authors to use a non-parametric approach in their study due to small dataset in the derivation of the standardized index or to use a Box-Cox transformation to resemble the normal distribution. Please address this issue in the revised manuscript.

4. Correlation analysis between standardized runoff index and geomorphologic indices. Please provide evidence that the employed indices follow the normal distribution. This must be demonstrated in the revised manuscript (a correlation matrix could be useful on this). If this is not the case different evaluation techniques could be followed (e.g. nonlinear techniques based on the mutual information and/or partial mutual information , non-parametric statistical tests (Kendall's tau and Spearman's rho rank

correlation coefficients could be used as alternatives of the Pearson coefficient).

5. since several frameworks have been developed to account nonstationarity (like the Generalized Additive Models for Location, Scale and Shape parameters, GAMLSS) could be included in the revised manuscript for comparison purposes. I would suggest to the authors to use their models with linear, quadratic and cubic terms in time to demonstrate that the employed models are appropriate and could be used subsequently in the simulation experiments.

Minor Comments

6. Line 95 - Previous works of the authors. The authors should explain in detail the novelty of this study in comparison with the previous works of the authors. A paragraph explaining the differences from these previous works should be included in the manuscript.

7. Lines 97-98. Correct the reference in the manuscript "Feng, 1997 & 1997".

8. Line 169. Correct the reference "Feng et al., 1997" in the text or in the bibliography list. There is not Feng et al., 1997 in the reference list.

References:

Van Loon, A. F. (2015), Hydrological drought explained. WIREs Water, 2: 359-392. doi:10.1002/wat2.1085.

Vasiliades, L., Loukas, A. & Liberis, N. (2011), A water balance derived drought index for Pinios river basin, Greece. Water Resour Manage 25: 1087-1101. https://doi.org/10.1007/s11269-010-9665-1.

For the motivations listed above, the paper in its present form needs revisions in order to evaluate the innovative character of the proposed method. The paper is of general interest for international audience and merits publication in NHESS journal when the major revisions and comments are addressed. Addressing these comments will

improve the quality of the paper and help the general reader of the paper.

---

## Referee Comment (RC2) · Anonymous Referee #2 · 5 Jun 2018

The manuscript "Study on the Driving Mechanism of Hydrologic Drought in Karst Basin Based on Landform Index: A Case Study of Guizhou, China," by He et al., attempts to describe the influences of topographic factors and characteristics of drought (severity, variability) on hydrologic drought in a karst basin of the Guizhou Province in southwest China.

My overall impression of the manuscript is that it is greatly hindered by a lack of clarity and innumerable issues of spelling, tense, punctuation and syntax, to such an extent that it is difficult to evaluate the paper and its potential contribution to the literature. Karst basins are in great need of examination, particularly as it pertains to drought,

which affects environmental and economic factors, as stated in this manuscript. However, after thoroughly reading and evaluating the manuscript, I suggest that it not be accepted for publication without addressing the language, syntax (incomplete sentences, tense disagreement) and punctuation issues and incoherence of thought. After these changes are made I suggest the authors resubmit as a new manuscript. While such matters are normally minor problems at this stage of the manuscript, the ubiquity of these issues in this case are a great deterrent to understanding the authors' meaning and presenting the ideas of the study and its conclusions. I appreciate the work of the authors and the analysis that they attempt to examine but I do not believe there is enough clarity at this point in the manuscript to determine if it truly is a contribution to field. I regret that this is the case and sincerely hope that a re-examination of the paper will resolve these issues so that it can be reconsidered for publication at another time. Below are some concerns worth mentioning which may improve the manuscript, starting with the problems in language throughout the paper, followed by some general comments, then some line-specific suggestions.

Regarding syntax, spelling, spacing, punctuation and consistency, in general the entire paper must be re-read and corrected for these problems. It would also benefit, in my opinion, from an English translator/editor who may help greatly with syntax. Many problems that were found in the citations and bibliography may be resolved through the bibliography and citation editor or the software used for the references. Almost every in-line and bibliography reference has issues with spacing and capitalization.

Check and resolve inconsistencies with capitalization, for instance, "Karst" vs "karst." To my knowledge there is not need for capitalization. Refer to "study" rather than "paper."

The use of italics and quotes is unnecessary in describing, for instance, "3 stages and 4 periods" of the droughts of study between 2000-2010, nor is it necessary or helpful to continue this usage for, say, Section 4.2.2. in which you also use quotes around "severe in the south and light in the north...," "Southwest Guizhou Province," "Zunyi

Area," "linear," "logarithmic," etc. Quotation marks such as these trivialize the findings and are greatly overused and distracting throughout the paper. Starting in Section 4.3.2. and continuing in 4.3.3, the subtitles, such as "(1) The driven by landform types" do not make sense in English. These all need to be resolved.

ABSTRACT

You first state that there is no significant impact of landforms on hydrological droughts but you did find that there was a significant impact of high-medium mountains, deep-high hills and high basins on hydrological droughts. This was a clear finding in your study and should be emphasized more in the abstract as well as in your conclusions.

INTRODUCTION

In general, this section would benefit from more clarity and order. It seems very loosely construed with a lot of citations but little coherence, that is, a lot of previous research listed but no real summary of actual findings and what needs to be done (justifying present study).

Line 35 Briefly explain more about differences in meteorological, agricultural, hydrologic and socio-economic drought.

Line 56 Was this in karst? Chalk? STUDY AREAS

Generally speaking, this section may be improved with a stratigraphic summary describing extent of karst, confinement, etc.

Line 106 "developed" karst? Does this mean mature karst development or developed for agriculture?

Line 110 "no heat in the summer?" Leave out the subjective general statements and just list rainfall, temps. . .

Line 111-113 Incomplete sentence. Also, quantify "poor lighting conditions"

DATA AND METHODS

Line 146-147 Incomplete sentence

Line 155 Unclear what normal value means

Line 180 "some processes?" do you mean analyses?

Line 186 Suggest introducing and describing the divisions of classifications of landforms for this study in the text of the methods as opposed to just in the tables RESULTS AND ANALYSIS Lines 257-259

There are several incomplete sentences here in a row making this incoherent. Line 337 "relative lightly areas of hydrologic droughts" does not make sense

Lines 345-346 Incomplete thought

Line 397 Suggest "denuded" as opposed to "naked"

Section 4.3.3, lines 388-422 requires a lot of corrections for tense, readability, coherence. CONCLUSIONS

Lines 455-458

Again, emphasize your significant findings regarding specific landforms instead of saying you didn't find any then saying you did.

REFERENCES

Needs work on almost every reference for spacing and capitalization in particular.

---

## Author Comment (AC1) · 10 Jun 2018

I have revised the paper according to the Reviewers' comments. For more, please see "the Blue Font Sections" on the revised manuscript.Thanks.

Please also note the supplement to this comment:
https://www.nat-hazards-earth-syst-sci-discuss.net/nhess-2018-10/nhess-2018-10-AC1-supplement.zip

---

## Author Comment (AC3) · 10 Jun 2018

First of all, thank the expert for his good opinions on my manuscript.There are some grammar questions raised by the expert about this manuscript,and I will further modify it.Thanks.